# Synthetic Aperture Radar Image Despeckling Based on Multi-Weighted Sparse Coding

**DOI:** 10.3390/e24010096

**Published:** 2022-01-07

**Authors:** Shujun Liu, Ningjie Pu, Jianxin Cao, Kui Zhang

**Affiliations:** School of Microelectronics and Communication Engineering, Chongqing University, Chongqing 400044, China; 201912021019@cqu.edu.cn (N.P.); jianxicao@126.com (J.C.); zk@cqu.edu.cn (K.Z.)

**Keywords:** synthetic aperture radar, image despeckling, nonlocal similarity, coefficient weighting, dictionary learning

## Abstract

Synthetic aperture radar (SAR) images are inherently degraded by speckle noise caused by coherent imaging, which may affect the performance of the subsequent image analysis task. To resolve this problem, this article proposes an integrated SAR image despeckling model based on dictionary learning and multi-weighted sparse coding. First, the dictionary is trained by groups composed of similar image patches, which have the same structural features. An effective orthogonal dictionary with high sparse representation ability is realized by introducing a properly tight frame. Furthermore, the data-fidelity term and regularization terms are constrained by weighting factors. The weighted sparse representation model not only fully utilizes the interblock relevance but also reflects the importance of various structural groups in despeckling processing. The proposed model is implemented with fast and effective solving steps that simultaneously perform orthogonal dictionary learning, weight parameter updating, sparse coding, and image reconstruction. The solving steps are designed using the alternative minimization method. Finally, the speckles are further suppressed by iterative regularization methods. In a comparison study with existing methods, our method demonstrated state-of-the-art performance in suppressing speckle noise and protecting the image texture details.

## 1. Introduction

Synthetic aperture radar (SAR) systems provide fine-resolution images regardless of the weather conditions. Therefore, they are widely used in environmental monitoring, military reconnaissance, and other fields [1,2,3]. Nevertheless, coherent imaging commonly causes multiplicative speckle noise in SAR images, which limits the analysis and interpretation of the scene and reduces the visualization of the images [4,5,6]. To avoid these problems, the speckle noise should be reduced as far as possible while retaining the structural features and texture information in SAR images [7,8,9,10,11].

In recent work, much attention was devoted to spatial-domain adaptive filtering based on Bayesian statistical estimation. Techniques such as linear minimum mean-square error (LMMSE)-based filtering [12] and maximum posterior probability (MAP)-based filtering [13] were introduced at this time. LMMSE filtering is performed by Lee [14], Frost [15], and Kuan filters [16], whereas MAP filtering [17] employs Gaussian and gamma filters. Although filters are easily implemented and have low computational complexity, they lose much of the texture information [18]. Despeckling methods in the wavelet domain have also been proposed. Initially, multiplicative noise was converted to additive noise through homomorphic filtering. Later, the normal inverse Gaussian model was introduced [19], which removes speckle noise without the shortcomings of spatial filter wave technology and has set a new direction for SAR image processing. Transform-domain-based methods better preserve the edges than spatial-domain filters, but tend to generate artifacts. Another important property of SAR images is nonlocal self-similarity, which is easily incorporated in SAR image despeckling. For example, nonlocal means (NLM) filtering [20] exploits self-similarity for image denoising. In an NLM filter, the resemblance between image patches is measured by the Euclidian distance and weighted averaging is performed in nonlocal regions. Inspired by the success of the NLM denoising filter, the authors of [21] developed the probabilistic patch-based (PPB) algorithm, in which the similarity between image patches in the SAR image is determined by the noise distribution rather than the traditional Euclidian distance.

In recent years, the continuous development of sparse theory has sparked interest in image denoising methods based on sparse representation [22,23,24,25,26]. In these methods, the most critical steps are selecting the appropriate dictionary, constructing the sparse model, and estimating the sparse coefficient. The earliest sparse representation methods, such as the discrete cosine transform and wavelet transform, relied on fixed, over-complete dictionaries. However, fixed dictionaries lack sufficient flexibility and self-adaptability for disparate image structures and the exact expression of signals. A more flexible dictionary would better characterize the prior information of the images. Recently, image patches have been identified as a flexible sparse target. Unlike the whole image, a patch is low-dimensional and overlapped patches provide opportunities for adaptively learning a dictionary or transforming the image into a sparse representation. Sparse dictionaries of training image patches extracted from images are known to ensure sparse representation coefficients for specific image structures [27,28]. The famous K singular value decomposition (K-SVD) algorithm [29] assumes that each patch has a sparse representation in a global over-complete dictionary. Each atom in the dictionary is then determined by minimizing the fitting error of the represented patches under the sparsity constraint. Considering the similarity and geometric direction of image patches, the authors of [30] classified image patches and learned an orthogonal dictionary for each type of image patch. As this method sparingly represents the types of image patches, it achieves faster dictionary learning, sparser coding, and better image reconstruction quality than the K-SVD algorithm. Unlike fixed dictionaries, a learned dictionary can adapt to the structural features of images and effectively obtains the sparse representations of images.

In recent works, the sparsity and self-similarity of images are often combined to improve the representation performance. The BM3D algorithm [31] combines similar image patches with three-dimensional (3D) transformation to enhance the sparsity level of 3D groups. For this purpose, BM3D performs a 3D wavelet transform on the groups and estimates the real-image wavelet coefficients by wiener filtering, which effectively removes Gaussian white noise from the images. The SAR-BM3D algorithm [32] extends the BM3D algorithm to SAR images. This algorithm replaces the wiener filter with a local LMMSE estimator that more accurately estimates the wavelet coefficients of the image, and hence reduces the speckle effect on SAR images. The fast adaptive nonlocal SAR despeckling (FANS) algorithm [33] is a faster and spatially-adaptive version of SAR-BM3D, which ensures a better speckle rejection in homogeneous areas. Dong et al. [34] proposed a nonlocal centralized sparse representation model based on the traditional sparse representation model, which estimates the sparse coefficients by nonlocal averaging of the image patches. As the image patches are used for constructing the coefficient correction terms in the sparse representation, the sparse coefficient is close to the real-image coefficient. To estimate the real image, some researchers exploit the low-rank property of similar image patch sets [35] and apply a soft thresholding that handles the singular values of similar noisy image patch sets.

As is well known, image regions differ in structure, so the sparsities of various image regions differ in the transformation domain. For this reason, imposing constant constraints on the coefficients is unlikely to fully utilize the prior information. To characterize the structural differences in the image, weighted processing of the image coefficients is required. The authors of [36] proposed a weighted nuclear norm minimization (WNNM) algorithm. The standard nuclear norm minimization (NNM) algorithm treats all singular values equally and shrinks them with the same threshold. However, the equalization process of NNM ignores the differences between image singular values in collections of similar image patches with different image prior information. The WNNM algorithm treats different singular values by assigning them with different weights. The weighting fully utilizes the prior information in different singular values, thus improving the flexibility of the kernel parameter constraint. Inspired by the Schatten *p*-norm and WNNM, the authors of [37] developed the weighted Schatten *p*-norm minimization model, which replaces the nuclear norm in the standard NNM model with the Schatten *p*-norm. They also recommended a weight parameter for the Schatten *p*-norm that obtains a new low-rank regular term. The problem is then solved by a generalized threshold contraction method. The trilateral weighted sparse coding model [38] has achieved good results in practical image denoising. This model accurately locates different noise intensities in different image regions, introduces three weight matrices into the model, and characterizes the statistics of realistic noise and image priors with data fidelity term and regularization term of the sparse coding framework. The high significance of the weights ensures that the estimated sparse coding coefficients approach the real-image coefficients.

Inspired by the above weighting idea, we propose a novel SAR image despeckling model based on dictionary learning and multi-weighted sparse coding. In the proposed multi-weighted sparse coding (MWSC), the efficient representation capacity of the orthogonal dictionary is integrated with the strong sparsity constraint of the multi-weighting model. This work makes three main contributions to the existing literature. First, we establish our general analysis dictionary learning and MWSC-based SAR image despeckling model. The proposed model continuously and efficiently trains the orthogonal dictionaries through SVD. Second, we introduce a weight matrix for nonuniform regularization, which explicitly and reasonably characterizes the diverse nonlocal characteristic of a group rather than performing uniform nonlocal regularizations. Another weight matrix is introduced to the data-fidelity term for characterizing the noise property. Third, we design an alternative minimization solution step for simultaneous orthogonal dictionary learning, weight parameter update, nonconvex sparse coding, and image reconstruction. As demonstrated in subsequent experimental results, the orthogonal dictionary learning and weight constraints both contribute to the final performance improvements. In terms of speckle noise removal and preservation of details, the MWSC recovered higher-quality images than other SAR image despeckling methods.

## 2. Traditional Sparse Coding Model

Speckles in SAR images can be characterized by the following multiplicative noise model [39]:(1)y=xu,
where y is the observed SAR image, x is the real value of the SAR image, and u is the fully developed speckle noise. First, the multiplicative noise model must be transformed into an additive noise model. To this end, the SAR image is preprocessed as
(2)y=xu=x+x(u−1)=x+n,
where n is the observed additive noise, which depends on the signal. In sparse representation theory, the sparse coding problem of each patch $y_{k}$ over dictionary D is formulated as
(3)$$c_{k}=\underset{c_{k}}{\arg\min}\frac{1}{2}{\left\|y_{k}-Dc_{k}\right\|}_{2}^{2}+\lambda {\left\|c_{k}\right\|}_{P},$$
where ${\left\|y_{k}-Dc_{k}\right\|}_{2}^{2}$ is the data-fidelity term, $y_{k}=R_{k}y$ is the image patch, and $R_{k}$ is a defined operator that extracts patch $y_{k}$ from y. ${\left\|c_{k}\right\|}_{P}$ is the regularization term denoting the image prior, and λ is the regularization parameter used for balancing the relationship between the sparse approximation error and the sparse constraint. The sparse coding process of each patch $y_{k}$ attempts to find a sparse coefficient $c_{k}$ over a given dictionary D. Therefore, most of the entries in the vector $c_{k}$ are zero or close to zero. Once $c_{k}$ is computed, the latent clean patch matrix ${\hat{x}}_{k}$ is estimated as ${\hat{x}}_{k}=Dc_{k}$. The clean image x to be reconstructed is then sparsely represented by a set of sparse codes {$c_{k}$} formulated as follows:(4)$$x=(\sum_{k=1}^{n}R_{k}^{T}R_{k}^{}{)}^{-1}\sum_{k=1}^{n}\left(R_{k}^{T}Dc_{k}\right)$$

## 3. The Proposed SAR Despeckling Algorithm

### 3.1. Weighted Sparse Representation Model

As the observed image y is seriously damaged by speckle noise, the image patch-based sparse coding method exploits only the local sparsity, which is insufficient for estimating the sparse coefficients $c_{k}$ from y. The correlation among image patches is another important prior information of the image. To exploit this prior, we construct and utilize groups of patches (rather than a single patch) as the basic unit of sparse representation. To retain the correlation in the group, regularizing its group sparsity is an obvious solution; however, the group sparsity may not be sufficiently guaranteed for all groups, because the levels of similarity widely vary in different image regions. This variability is demonstrated on the “Lena” image in Figure 1. Two groups, labeled “A” and “B” in Figure 1, are found in the homogeneous region and a heterogeneous region, respectively. The enlarged patches in Figure 1b clarify the similarity differences between the groups “A” and “B.” In the current group-based sparse coding approaches, nonlocal similarity is enforced by uniform nonlocal regularizations with the same penalty parameter, such as the norm. If the same constraints are imposed on each image patch, the structural features of the image cannot be fully captured and utilized, and some details and texture information are inevitably lost from the image. In addition, the data-fidelity term in the sparse coding model ensures the authenticity of the sparse coefficient in the sparse representation process and conforms to the image despeckling process. In reality, the additive noise n depends on the signals in the SAR image and its level varies in different local patches. Therefore, in the actual SAR image despeckling process, the important work of the data-fidelity term ${\left\|y_{k}-Dc_{k}\right\|}_{2}^{2}$ is varying the local noise intensity.

To rectify the problems in traditional patch-based sparse representation, we build groups of similar patches and simultaneously exploit the local sparsity and the nonlocal self-similarity of SAR images in a unified framework. We first divide the image y into n overlapped patches, denoting each patch by a vector $y_{i}$, i.e., i=1,2,…,n. For each target image patch $y_{i}$ in the S×S training window, we search for N−1 best matched patches. Finally, all patches (the target image patch and all similar image patches) are stacked into a matrix $Y_{i}$ representing the group. This step requires a suitable metric for measuring the similarity between noisy patches. Here, the Euclidean distance is selected as the similarity metric between the target image patch $y_{i}$ and the candidate patch $y_{i,j}$ in the training window. The Euclidean metric is given by
(5)$$d(y_{i},y_{i,j})={\left\|y_{i}-y_{i,j}\right\|}_{2}^{2}.$$

The group $Y_{i}$ containing the searched similar patches is then represented as
(6)$$Y_{i}=[R_{i,1}y,R_{i,2}y,\ldots ,R_{i,N}y]=[y_{i,1},y_{i,2},\ldots y_{i,N}],$$
where $R_{i,j}$ is an operator that extracts the *j*-th similar patch $y_{i,j}=R_{i,j}y$ from y, and N is the number of similar patches in group $Y_{i}$. As all image patches in $Y_{i}$ share a similar underlying structure, the column vectors of $Y_{i}$ are strongly correlated. The sparse coefficients of the internal patches of group $Y_{i}$ are collected into a coefficient matrix denoted by ***C*** = [***c***_1_, ***c***_2_,…,***c****_N_*].

The current group-based SAR image despeckling approaches impose the same constraint on each group. Because they ignore the varying local noise intensity, they cannot fully capture and utilize the nonlocal characteristics of the image. To improve the accuracy of C, we analyze its probability statistics. More specifically, we apply the maximum a posterior (MAP) estimation derived from Gaussian Naïve Bayesian formula to the coefficient matrix C:(7)$$\begin{array}{ll} C=\underset{C}{\arg\max}\;P(C|\;Y_{i}) & =\underset{C}{\arg\max}\;\frac{P(Y_{i}|\;C)P(C)}{P(Y_{i})}\\ & =\underset{C}{\arg\max}\;P(Y_{i}|\;C)P(C)\\ & =\underset{C}{\arg\max}\{\;\ln\;P(Y_{i}|\;C)+\ln P(C)\} \end{array}$$

In Equation (7), $\ln\;P(Y_{i}|\;C)$ is the data-fidelity term. Its form depends on the statistical characteristics of the image noise that penalizes the differences between group $Y_{i}$ and the coefficient matrix C. As the noise intensity varies in different regions of the SAR image, we assume that the additive noise n follows a Gaussian distribution of noise standard deviations. Hence, the data-fidelity term $P(Y_{i}|\;C)$ is defined as
(8)$$P(Y_{i}|\;C)=\prod_{j=1}^{N}\frac{1}{\sqrt{2\pi }\sigma_{j}}\exp(-\frac{1}{2\sigma_{j}^{2}}{\left\|y_{i,j}-Dc_{j}\right\|}_{2}^{2}),$$
where $\sigma_{j}$ is the noise standard deviation of the additive noise in image patch $y_{i,j}$.

By the group property, all image patches in $Y_{i}$ share a similar underlying structure. Therefore, their coefficient vectors under one dictionary should share the same sparse profile. Specifically, the elements in one row of C have the same magnitude. We thus assume that each row of C is independent of all other rows and that its scale parameters follow a Laplace distribution. Therefore, the prior term in Equation (7) is defined as
(9)$$P(C)=\prod_{j=1}^{N}\prod_{l=1}^{M}\frac{1}{2\delta_{l}}\exp(-\frac{\left|c_{j}^{l}\right|}{\delta_{l}})=\prod_{l=1}^{M}\frac{1}{{\left(2\delta_{l}\right)}^{N}}\exp(-\frac{{\left\|c^{l}\right\|}_{1}}{\delta_{l}}),$$
where $c^{l}$ is denotes the *l*-th row vector of C and $\delta_{l}$ are the scale parameters of $c^{l}$. Substituting Equations (8) and (9) into Equation (7), the MAP estimation model of coefficient matrix C is obtained as
(10)$$\begin{array}{ll} C & =\underset{C}{\arg\min}\sum_{j=1}^{N}{\left\|\sqrt{2}(y_{i,j}-Dc_{j})\sigma_{i,j}^{-1}\right\|}_{F}^{2}+\sum_{l=1}^{M}{\left\|\delta_{l}{}^{-1}c^{l}\right\|}_{1}\\ & =\underset{C}{\arg\min}{\left\|(Y_{i}-DC)Q_{1}\right\|}_{F}^{2}+{\left\|Q_{2}^{-1}C\right\|}_{1} \end{array}$$
where $Q_{1}=diag(\sqrt{2}\sigma_{1}^{-1},\ldots ,\sqrt{2}\sigma_{N}^{-1})$ and $Q_{2}=diag(\delta_{1},\ldots ,\delta_{M})$ are two diagonal matrices. The design of $Q_{1}$ and $Q_{2}$ to characterize the varying statistics of realistic noise and the sparsity prior of SAR images is described in Section 3.2.

Unlike the traditional sparse representation model, we modify the data-fidelity term and regularization term with the weight matrices $Q_{1}$ and $Q_{2}$, respectively. The weight matrix $Q_{1}$ characterizes the residual $Y_{i}-DC$ in the data-fidelity term ${\left\|(Y_{i}-DC)Q_{1}\right\|}_{F}^{2}$. When the diagonal element $\sqrt{2}\sigma_{j}^{-1}$ is large, the speckle noise intensity is high and a stronger constraint is imposed on the weighted residual term. Conversely, when the speckle noise intensity is low, the constraint on the residual term is weakened. By introducing the weight matrix $Q_{1}$ into the data-fidelity term, we capture and utilize the realistic noise properties in different patches. Meanwhile, $Q_{2}$ in the regularization term reflects the importance of each element in C. Specifically, one weight in $Q_{2}$ is assigned to the elements in one row of C to enforce the same sparse profile among similar patches, whereas the elements of C in different rows are assigned different weights that change the strengths of the sparsity constraints. Overall, the weighted sparse constraint model not only reflects the varying local noise intensity, but also simultaneously utilizes the patch-level sparsity and relevance among patches, which effectively improves the adaptability of the model to real SAR images and the accuracy of the coefficient matrix C.

### 3.2. Adaptive Matrix Parameter Learning

Because the noise intensity varies in different image regions, the weight matrix $Q_{1}$ must learn to adaptively describe the statistical properties of the noise. In the actual speckle-reduction process, the weight matrix $Q_{1}$ is updated through iterative regularization. During the *k*-th iteration, the parameter $\sigma_{j}$ in weight $Q_{1}$ is updated as
(11)$$\sigma_{j}=\gamma \sqrt{abs(\sigma^{2}-{\left\|y_{j}-y_{j}^{k}\right\|}_{2}^{2}/N)},$$
where σ is the noise benchmark parameter, γ<1 is a control factor, which is tuned according to the intensity of noise. In general, a smaller γ is set for low noise level and a larger γ is set for high noise level. $y_{j}$ is the *j*-th image patch extracted from the original observation image y, and $y_{j}^{k}$ is the *j*-th image patch extracted from the input image $y^{k}$ during the *k*-th iteration of iterative regularization (see Section 3.4. for details).

An effective regularization term that describes the image priors is required for a high-quality recovered image. In the proposed model given by Equation (10), the combinational regularization term ${\left\|Q_{2}^{-1}C\right\|}_{1}$ is precisely controlled by a weight matrix $Q_{2}$ describing the importance of the different elements of the coefficient matrix. In our method, the weight matrix $Q_{2}$ of the regularization term is learned from $Y_{i}$ for adaptively adjusting the weights of the coefficients. Clearly, $Q_{2}$ is individual to group $Y_{i}$ and effectively captures the specific structural information of that group.

Fixed dictionaries are unlikely to effectively express the signals in a sparse representation, but a trained dictionary is expected to enhance the sparsity of coefficients. The dictionary learning method in the traditional sparse coding model (Equation (3)) jointly optimizes the dictionary D and the coefficient vector $c_{k}$. The dictionary D learned by this method adapts to a given whole image, not merely a group. When the same dictionary is shared by all image patches, it may not adapt to specific local structures in the image. Instead of learning a redundant dictionary for the whole image, we directly learn an individual dictionary D for each group $Y_{i}$, which covers all patches in that group. The model also imposes a tight frame constraint $D^{T}D=I$, which not only restrains the correlations among dictionary atoms, but also simplifies the complexity of the sparse decomposition operations.

Finally, the complete group-based dictionary learning and MWSC model is defined as follows:(12)$$(C,Q_{2},D)=\underset{C,Q_{2},D}{\arg\min}{\left\|(Y_{i}-DC)Q_{1}\right\|}_{F}^{2}+{\left\|Q_{2}^{-1}C\right\|}_{1}\;s.t.\;D^{T}D=I,$$
where C, $Q_{2}$ and D are matrices to be solved.

### 3.3. Model Optimization

To facilitate the solution of Equation (12), we define an auxiliary variable $A=Q_{2}^{-1}C$. Equation (12) is then rewritten as
(13)$$(A,Q_{2},D)=\underset{A,Q_{2},D}{\arg\min}{\left\|(Y_{i}-DQ_{2}A)Q_{1}\right\|}_{F}^{2}+{\left\|A\right\|}_{1}\;s.t.\;D^{T}D=I.$$

Equation (13) includes three unknown variables. Because Equation (13) is nonconvex, it is solved by an alternative minimization method. In each iteration, Equation (13) is decomposed into three subproblems: A, $Q_{2}$ and D for learning the orthogonal dictionary D, updating the weight $Q_{2}$, and sparse coding of A, respectively.

Given D and A, the subproblem of Equation (13) with respect to $Q_{2}$ is
(14)$$Q_{2}=\underset{Q_{2}}{\arg\min}{\left\|(Y_{i}-DQ_{2}A)Q_{1}\right\|}_{F}^{2}\;.$$

As D is an orthogonal dictionary, the subproblem of $Q_{2}$ can be rewritten as
(15)$$\underset{}{Q_{2}=}\underset{Q_{2}}{\arg\min}{\left\|D^{T}Y_{i}Q_{1}-Q_{2}AQ_{1}\right\|}_{F}^{2}.$$

The objective function Equation (15) is separable into the diagonal elements of $Q_{2}$ and its minimization problem can be consequently decoupled into multiple independent scalar optimizations of the form
(16)$$\underset{\delta_{l}}{\delta_{l}=\arg\min}{\left\|{\left(D^{T}Y_{i}Q_{1}\right)}_{l}-\delta_{l}{\left(AQ_{1}\right)}_{l}\right\|}_{F}^{2},$$
where ${\left(D^{T}Y_{i}Q_{1}\right)}_{l}$ and ${\left(AQ_{1}\right)}_{l}$ represent the *l*-th rows of the matrices $D^{T}Y_{i}Q_{1}$ and $AQ_{1}$, respectively. Equation (16) is a quadratic minimization problem, so its diagonal element $\delta_{l}$ is solved as
(17)$$\delta_{l}=\frac{{\left(D^{T}Y_{i}Q_{1}\right)}_{l}{\left(AQ_{1}\right)}_{l}^{T}}{{\left\|{\left(AQ_{1}\right)}_{l}\right\|}_{2}^{2}}.$$

Estimating each *l*-th diagonal element $\delta_{l}$ of $Q_{2}$ by Equation (17), the solution of $Q_{2}$ is obtained.

Given $Q_{2}$ and A, the subproblem of Equation (13) with respect to D is
(18)$$\underset{D}{D=\arg\min}{\left\|Y_{i}Q_{1}-DQ_{2}AQ_{1}\right\|}_{F}^{2}\;s.t.\;D^{T}D=I.$$

Equation (18) is a Frobenius norm minimization problem with an orthogonal constraint, which can be solved by SVD of $Y_{i}Q_{1}{\left(Q_{2}AQ_{1}\right)}^{T}$ as
(19)$$U\Delta V^{T}=Y_{i}Q_{1}{\left(Q_{2}AQ_{1}\right)}^{T},$$
where U and V are the matrices of the left and right singular vectors, respectively, and Δ is the singular value matrix. By SVD, Equation (19) is solved as
(20)$$D=UV^{T}.$$

Clearly, the dictionary D is self-adaptive to each group $Y_{i}$ and requires only one SVD for each group, which is computationally efficient.

Given D and $Q_{2}$, the subproblem of Equation (13) with respect to **A** is
(21)$$A=\underset{A}{\arg\min}{\left\|(Y_{i}-DQ_{2}A)Q_{1}\right\|}_{F}^{2}+{\left\|A\right\|}_{1}.$$

As D is orthogonal, the first Frobenius norm term in Equation (21) is rewritten as ${\left\|D^{T}Y_{i}Q_{1}-Q_{2}AQ_{1}\right\|}_{F}^{2}$. The minimization of Equation (21) is separable into the elements of the matrices A and $Y_{i}$ as
(22)$$a_{j}=\underset{a_{j}}{\arg\;\min}\frac{2}{\sigma_{j}^{2}}{\left\|D^{T}y_{i,j}-Q_{2}a_{j}\right\|}_{F}^{2}+{\left\|a_{j}\right\|}_{1}.$$
where $a_{j}$ is the *j*-th column vector of matrix A. The minimization of Equation (22) can be further decoupled into multiple independent scalar optimizations of the form
(23)$$a_{j,l}=\underset{a_{j,l}}{\arg\min}\frac{1}{2}{\left(a_{j,l}-\frac{d_{l}^{T}y_{i,j}}{\delta_{l}}\right)}^{2}+\frac{\sigma_{j}^{2}}{4\delta_{l}^{2}}\left|a_{j,l}\right|,$$
where $a_{j,l}$ is the *l*-th element of $a_{j}$ and $d_{l}^{}$ is the *l*-th column of dictionary D. Equation (23) can be efficiently and accurately solved by soft thresholding, that is
(24)$$a_{j,l}=\mathrm{sign}(\frac{d_{l}^{T}y_{i,j}}{\delta_{l}})\max(\left|\frac{d_{l}^{T}y_{i,j}}{\delta_{l}}\right|-\frac{\sigma_{j}^{2}}{4\delta_{l}^{2}},0),$$
where sign(·) is a symbolic function. The soft-thresholding approach provides an explicit solution to the original Equation (21).

### 3.4. SAR Image Despeckling

Equations (14), (18) and (21) are solved until the stopping criterion is satisfied. The clean group $X_{i}$ is then estimated using the solved D, A and $Q_{2}$ as $X_{i}=DQ_{2}A$. Based on the estimated clean group $X_{i}$, the SAR image reconstruction model is constructed as
(25)$$x=\underset{x}{\arg\min}{\left\|x-y\right\|}_{2}^{2}+\eta \sum_{i}\sum_{j}{\left\|R_{i,j}x-{\left(DQ_{2}A\right)}_{j}\right\|}_{2}^{2},$$
where η is a regularization parameter and ${\left(DQ_{2}A\right)}_{j}$ is the *j*-th column of the estimated clean group $DQ_{2}A$. Equation (25) is a least squares problem that is minimized as follows:(26)$$x={\left(I+\eta \sum_{i}\sum_{j}R_{i,j}^{T}R_{i,j}\right)}^{-1}(\;y+\eta \sum_{i}\sum_{j}R_{i,j}^{T}{\left(DQ_{2}A\right)}_{j}),$$
where $\sum_{i}\sum_{j}R_{i,j}^{T}R_{i,j}$ is a diagonal matrix whose diagonal elements are the numbers of the corresponding pixels extracted from all similar patches. The inverse of the large-scale matrix $I+\eta \sum_{i}\sum_{j}R_{i,j}^{T}R_{i,j}$ is easily obtained. Solving Equation (26) yields the reconstructed SAR image.

To further improve the speckle-reduction performance of the proposed model, we repeatedly remove speckles using an iterative regularization technique that filters the noise back to the denoised image as the new input image:(27)$$y^{k+1}=x^{k}+\xi (y-x^{k}),$$
where k is the number of iterations and ξ is the relaxation parameter. The above despeckling process is repeated on the input image $y^{k+1}$ to reconstruct the SAR image $x^{k+1}$. This procedure is repeated several times until the number of iterations reaches the threshold M, and the final despeckled SAR image is output from the model. Figure 2 is a flowchart of the MWSC algorithm for suppressing speckle noise in SAR images.

## 4. Experimental Results and Analysis

In this section, we experimentally verify the performance of the proposed MWSC approach for SAR image despeckling. The despeckling performance was compared with those of Gamma-MAP filters, the Log-KSVD method, the PPB method, the POTDF method, the FANS method, and the SAR-BM3D method. PPB and SAR-BM3D are well-known classical algorithms for processing SAR images. The executable codes of the competing methods were downloaded from the authors’ website (http://www.grip.unina.it/research/80-sar-despeckling.html) (16 October 2021).

To investigate the impacts of patch size and overlapping factor, despeckling experiments on three test images are separately conducted under various patch sizes and overlapping factors. Figure 3 shows the curves of PSNR versus patch size and overlapping factor. From Figure 3a, we have the following observations. As the patch size increases, the PSNR first increases and then drops. This is because using a large patch size allows more robust discrimination between noisy patches. However, using a too large patch will prevent the algorithm from finding enough similar patches. From Figure 3b, it is concluded that the performance of our proposed algorithm is not quite sensitive to overlapping factor since all the curves are almost flat. The highest performance for each test image is achieved with the overlapping factor in the range [2,5]. Based on the above analysis, in the proposed MWSC algorithm, we set the size of image patch as 8 × 8 and overlapping factor is empirically set as 4. Besides, the range S×S of the training window for searching similar patches is set as 30 × 30. The number of image patches in one group is set as 32 and the size of the learning dictionary is set as 64 × 64.

As noise-free signals are lacking in real SAR images, we simulated a SAR image for the experiment based on an optical image. In our simulated test, we chose the “House” and “Cameraman” images of size 256 × 256 and the “Lena” image of size 512 × 512. The despeckling performance was quantitatively evaluated by two performance measures: the peak-signal-to-noise ratio (PSNR) and the structural similarity index (SSIM), which evaluate the speckle-suppression capability of the algorithm and the structural similarity of the recovered image to the original image, respectively.

On real SAR images, the despeckling performance was evaluated by the equivalent number of looks (ENL) and the overall edge preservation index (EPI). The ENL is a widely used index that measures the degree of speckle suppression in a homogeneous region. A large ENL denotes a strong speckle-suppression ability. Meanwhile, the EPI reflects the degree of image-detail preservation. A large EPI indicates a strong detail-preservation ability.

### 4.1. Despeckling Results of Simulated SAR Images

We first validated the proposed method on simulated SAR images generated by adding different levels of gamma speckle noise (*L* = 4 or 16) to the natural images “House,” “Cameraman”, and “Lena”. The number of looks *L* represents the speckle noise level, where smaller *L* denotes a greater noise intensity. The original images are shown in Figure 4.

Figure 5, Figure 6, Figure 7 and Figure 8 show the images obtained by the different despeckling methods on the simulated optical images contaminated by speckle. All methods achieved satisfactory despeckling performance, although Gamma-MAP introduced some obvious artifacts that severely degraded the image quality of Lena (Figure 5c, Figure 6c, Figure 7c and Figure 8c). The Log-KSVD method left significant noise residues in the smoothed areas (Figure 5b and Figure 6b). Although PPB provided a much smoother result than Log-KSVD and Gamma-MAP, it blurred some important details, such as the hat texture in Figure 6d and the window edge in Figure 7d. POTDF keeps working very well on the heterogeneous region while unsatisfactory on the homogeneous region (Figure 7e). The SAR-BM3D method well preserved the texture information of the image and reasonably suppressed the speckle noise, but some residual noise appeared in the despeckled image of Lena (Figure 6f). Likewise, FANS produces a smoother output, but many wavelet-related artifacts appear. We observe that the proposed MWSC model adequately suppressed speckle noise and better restored the edges and textures of the original images than the competing methods.

To demonstrate the processing results of each algorithm on image details and point targets, we enlarged some local details in Figure 5 and Figure 7. The enlargements are shown in Figure 9 and Figure 10, respectively. Here we compared only the results of PPB, POTDF, SAR-BM3D, FANS and MWSC, the results of the Log-KSVD and Gamma-MAP algorithms were omitted because their visual effects were poor. Visible speckle noise persisted in the House image despeckled by PPB (Figure 10). The POTDF tended to produce an unfavorable oversmoothing effect. On the other hand, SAR-BM3D, FANS, and the proposed MWSC method best preserved the image details, but MWSC better suppressed the speckle noise than the SAR-BM3D and FANS. For example, significant speckle noise appeared in the enlarged local details of House obtained by SAR-BM3D and FANS (Figure 10). Besides eliminating the speckle noise, the proposed MWSC algorithm better preserved the sharper edges and finer details than the other algorithms, and thereby obtained much clearer and better visual results. The corresponding PSNR and SSIM evaluations of the different algorithms on the three synthetic images are displayed in Table 1. The MWSC algorithm achieved the highest PSNR and SSIM results in most cases (highlighted in bold font), consistent with the visual inspections.

### 4.2. Despeckling Results of Real SAR Images

In this experiment, the despeckling performances of the proposed and existing algorithms were tested on real SAR images (see Figure 11). The despeckled results of the real SAR images obtained by the compared methods are shown in Figure 12, Figure 13 and Figure 14.

As shown in Figure 12, Figure 13 and Figure 14, the proposed MWSC outperformed its competitors on the real SAR images. The Gamma-MAP algorithm eliminated less speckle noise than other methods and introduced serious aliasing artifacts. It also blurred edges of the details, as shown in Figure 12c and Figure 13c.The Log-KSVD algorithm slightly improved the despeckling effect but retained some obvious speckle noise (see Figure 12b and Figure 13b). Although PPB effectively suppressed the speckle noise from the SAR images, it blurred some edges and strong targets Figure 12d and Figure 14d). From Figure 12e and Figure 13e, we can find that POTDF still remains much noise in both homogeneous and heterogeneous areas. SAR-BM3D and FANS preserve all structures very well, with an accuracy comparable to that of MWSC. The MWSC, however, succeed in removing speckle in both homogeneous and heterogeneous areas, providing a sharper result and contributing to a better perceived quality. To more intuitively compare the detailed processing of the various algorithms, Figure 15 and Figure 16 present local enlargements of the despeckled images obtained by the despeckling methods. Again, the proposed MWSC method achieved a higher visual quality than the other despeckling methods.

To quantitatively evaluate the despeckling results, the corresponding ENLs and EPIs of the three real images are provided in Table 2, with the best values highlighted in bold. The ENL values were calculated in homogeneous regions of the real SAR images. Regions 1 and 2 in Table 2 correspond to the red and yellow regions, respectively, in the SAR images (such as the Figure 12a). PPB and the proposed MWSC achieved higher ENL than the other despeckling methods, affirming that the PPB and MWSC algorithms have stronger speckle-reduction ability in the homogeneous regions than the remaining algorithms. This conclusion is consistent with the above visual inspection. Meanwhile, SAR-BM3D, FANS, and MWSC obtained higher EPI than PPB, Gamma-MAP, Log-KSVD and POTDF, confirming the superior performance of SAR-BM3D, FANS and MWSC in image-detail retention. In short, the proposed MWSC provided the best compromise between speckle reduction and detail preservation.

### 4.3. Impacts of Weights and Dictionaries on Performance

To better understand the roles of the weights $Q_{1}$ and $Q_{2}$ and the learning dictionary D in the MWSC model, we compared the performances of the proposed MWSC and several MWSC-based benchmark methods. We first replaced the weight $Q_{1}$ with the unit diagonal matrix, forming the QWSC benchmark model. Replacing the learning dictionary D with the wavelet dictionary, we then formed the DWSC benchmark model.

Figure 17 and Figure 18 present the despeckled images obtained by the different despeckling methods. Log-KSVD retained a large amount of speckle noise in the “House” image (Figure 17a). QWSC improved the visual effect of “House” because the weights $Q_{2}$ improve the sparse representation capability of the QWSC algorithm and further enhance the speckle-suppression performance (Figure 17b). MWSC achieved a higher visual effect than QWSC because the weights $Q_{1}$ effectively capture and utilize the statistical properties of the noise (Figure 18b,d). Although the result of DWSC exhibited no obvious speckle noise, a small amount of detailed information was lost (Figure 18c). Obviously, the learning dictionary in the proposed model adapted to different structures of the image to improve the speckle-reduction performance of the model.

Table 3 and Table 4 give the numerical evaluation results of the different algorithms on the “House” and real SAR2 images, respectively, with the best results marked in bold. As shown in Table 3, MWSC achieved higher PSNR and SSIM values than the benchmark algorithms on “House,” confirming that weight $Q_{1}$, weight $Q_{2}$, and dictionary learning all improved the PSNR and SSIM values of the recovered images to different degrees. Meanwhile, the ENL and EPI values in Table 4 clarify that the proposed BWCS algorithm well balanced the speckle noise suppression with detail preservation in the real SAR image, outperforming the Log-KSVD algorithm and the benchmark algorithms.

This comprehensive analysis demonstrates that introducing weight $Q_{1}$ improves the noise-statistical properties of SAR images, while introducing weight $Q_{2}$ captures the sparse a priori properties of SAR images. The learning dictionary is more adaptable to different image structures than the fixed dictionary, thereby enhancing the signal sparsity and achieving competitive SAR image-despeckling performance.

## 5. Conclusions

This paper proposed a multiweighted SAR image despeckling method based on the traditional sparse representation model and combined it into a framework that unifies dictionary learning and coefficient weighting. In this method, the structure group is constructed using the nonlocal similarity between SAR image patches and is employed as the processing object of speckle reduction in SAR images. This approach significantly improves the sparsity of the image coefficients. Borrowing the idea of coefficient weighting, we also designed a data-fidelity term for the total downscaling model and a regularization term to enhance the reliability and accuracy of image coefficient estimation. An orthogonal dictionary was trained on sets of similar image patches to elevate the sparsity of the coefficients and simplify the complexity of sparse coding. Throughout the iterations, the algorithm updates the weight matrix of each similar image patch set to improve the full-medium adaptation. By combining the statistical parameter estimation of SAR images with iterative regularization, we further improved the quality of the images during the solution process. In comparison experiments, the proposed method achieved comparable or better despeckling performance than four competing methods, in terms of speckle suppression, detail preservation, and visual effect. The results verified the reasonableness and effectiveness of the proposed method.

## Figures and Tables

**Figure 1 entropy-24-00096-f001:**
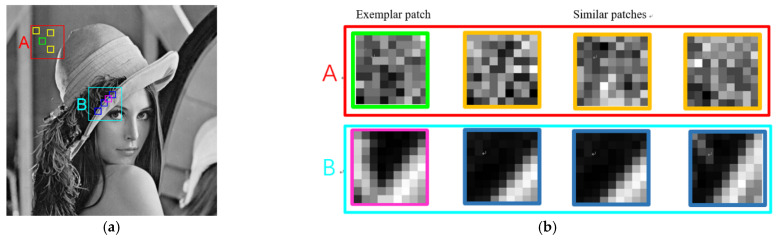
Comparison of similarity levels of two groups found in the “Lena” image. (**a**) Group A (in the red window) and Group B (in the cyan window) include exemplary and similar patches; (**b**) Zoomed-in exemplary and similar patches in groups “A” and “B”.

**Figure 2 entropy-24-00096-f002:**
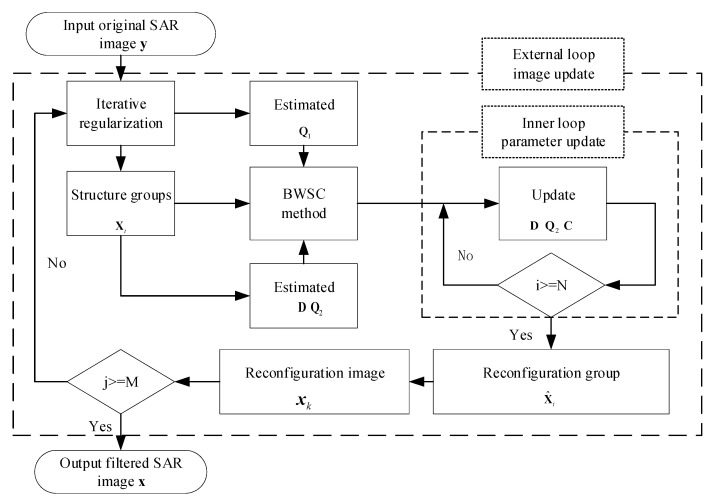
Flowchart of the MWSC algorithm for SAR image despeckling.

**Figure 3 entropy-24-00096-f003:**
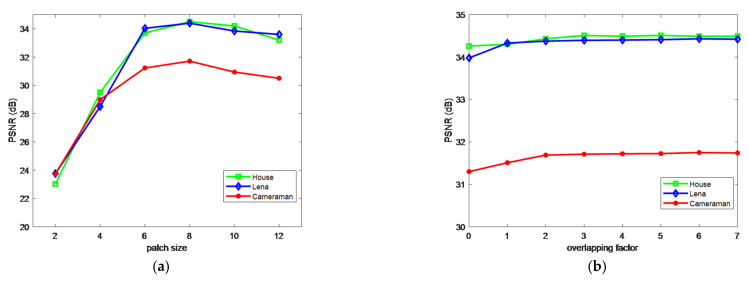
Performance comparison with various patch sizes (**a**) and overlapping factors (**b**) for three test images.

**Figure 4 entropy-24-00096-f004:**
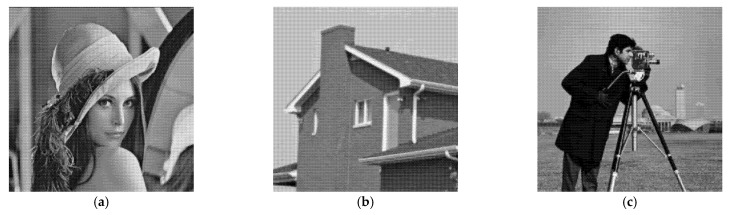
Original images used in the comparison experiments. (**a**) Lena (512 × 512); (**b**) House (256 × 256); (**c**) Cameraman (256 × 256).

**Figure 5 entropy-24-00096-f005:**
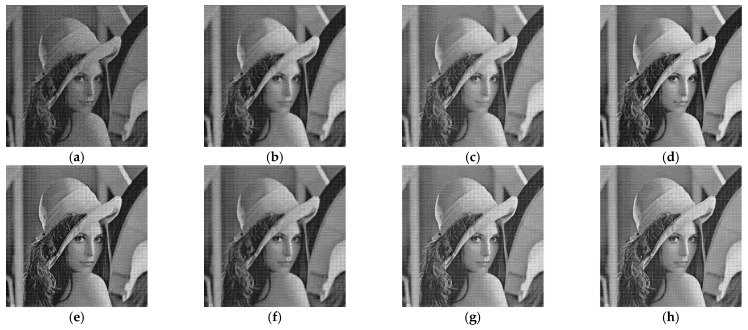
Despeckling results of Lena corrupted by 16-look speckle. (**a**) Noisy image; (**b**) Log-KSVD; (**c**) Gamma-MAP; (**d**) PPB; (**e**) POTDF; (**f**) SAR-BM3D; (**g**) FANS; (**h**) MWSC.

**Figure 6 entropy-24-00096-f006:**
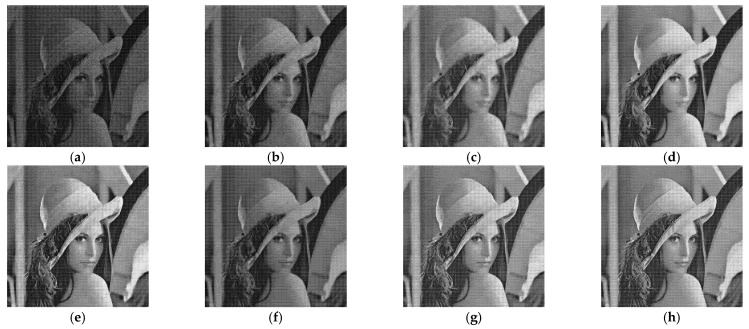
Despeckling results of Lena corrupted by 4-look speckle. (**a**) Noisy image; (**b**) Log-KSVD; (**c**) Gamma-MAP; (**d**) PPB; (**e**) POTDF; (**f**) SAR-BM3D; (**g**) FANS; (**h**) MWSC.

**Figure 7 entropy-24-00096-f007:**
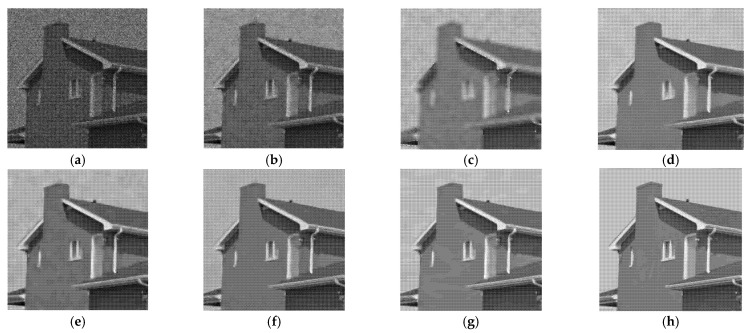
Despeckling results of House corrupted by 4-look speckle. (**a**) Noisy image; (**b**) Log-KSVD; (**c**) Gamma-MAP; (**d**) PPB; (**e**) POTDF; (**f**) SAR-BM3D; (**g**) FANS; (**h**) MWSC.

**Figure 8 entropy-24-00096-f008:**

Despeckling results of cameraman corrupted by 16-look speckle. (**a**) Noisy image; (**b**) Log-KSVD; (**c**) Gamma-MAP; (**d**) PPB; (**e**) POTDF; (**f**) SAR-BM3D; (**g**) FANS; (**h**) MWSC.

**Figure 9 entropy-24-00096-f009:**
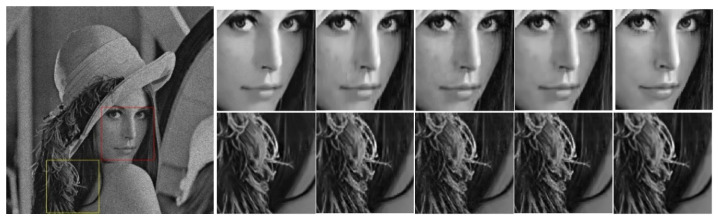
Enlarged details of the despeckling results on the Lena image with *L* = 16 (left to right: PPB algorithm, POTDF algorithm, SAR-BM3D algorithm, FANS algorithm, MWSC algorithm).

**Figure 10 entropy-24-00096-f010:**
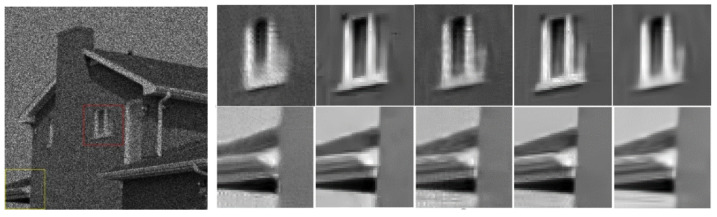
Enlarged details of the despeckling results on the House image with *L* = 4 (left to right: PPB algorithm, POTDF algorithm, SAR-BM3D algorithm, FANS algorithm, MWSC algorithm).

**Figure 11 entropy-24-00096-f011:**

Real SAR images used in the comparison experiments. (**a**) SAR1 (256 × 256); (**b**) SRA2 (256 × 256); (**c**) SAR3 (512 × 512).

**Figure 12 entropy-24-00096-f012:**
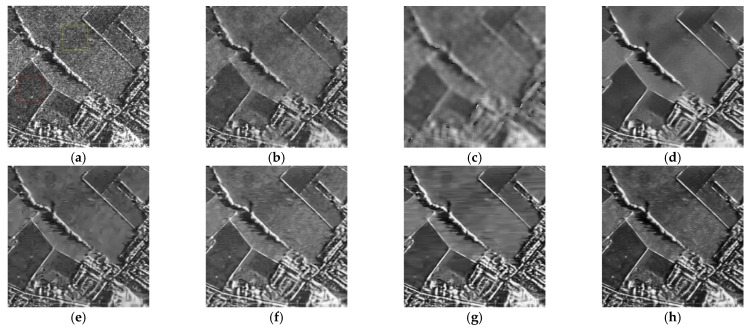
Despeckling results of the real SAR1 image. (**a**) Noisy image; (**b**) Log-KSVD; (**c**) Gamma-MAP; (**d**) PPB; (**e**) POTDF; (**f**) SAR-BM3D; (**g**) FANS; (**h**) MWSC.

**Figure 13 entropy-24-00096-f013:**
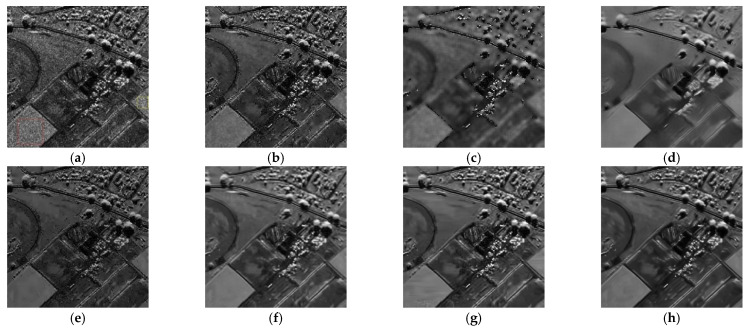
Despeckling results of the real SAR2 image. (**a**) Noisy image; (**b**) Log-KSVD; (**c**) Gamma-MAP; (**d**) PPB; (**e**) POTDF; (**f**) SAR-BM3D; (**g**) FANS; (**h**) MWSC.

**Figure 14 entropy-24-00096-f014:**

Despeckling results of the real SAR3 image. (**a**) Noisy image; (**b**) Log-KSVD; (**c**) Gamma-MAP; (**d**) PPB; (**e**) POTDF; (**f**) SAR-BM3D; (**g**) FANS; (**h**) MWSC.

**Figure 15 entropy-24-00096-f015:**
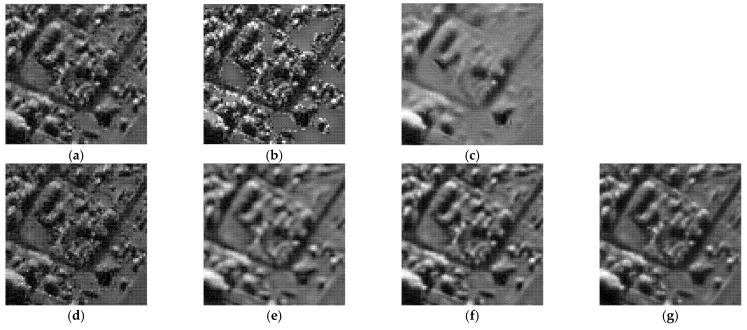
Despeckling results of a local enlargement of Figure 9. (**a**) Noisy image; (**b**) Log-KSVD; (**c**) Gamma-MAP; (**d**) PPB; (**e**) POTDF; (**f**) SAR-BM3D; (**g**) FANS.

**Figure 16 entropy-24-00096-f016:**
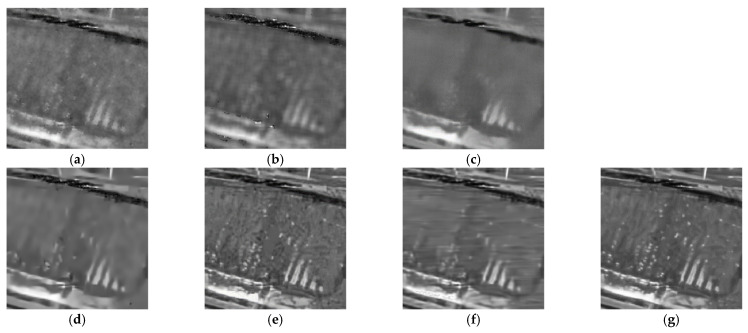
Depeckling results of a local enlargement of Figure 10. (**a**) Noisy image; (**b**) Log-KSVD; (**c**) Gamma-MAP; (**d**) PPB; (**e**) POTDF; (**f**) SAR-BM3D; (**g**) FANS.

**Figure 17 entropy-24-00096-f017:**
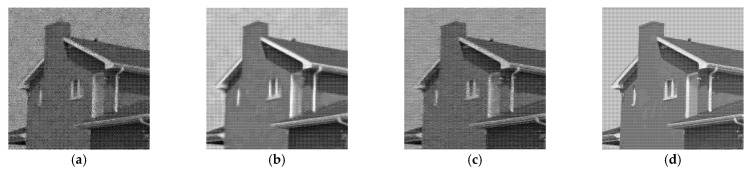
Despeckling results of House corrupted by 4-look speckle. (**a**) Log-KSVD; (**b**) QWSC; (**c**) DWSC; (**d**) MWSC.

**Figure 18 entropy-24-00096-f018:**

Despeckling results of real SAR2 image. (**a**) Log-KSVD; (**b**) QWSC; (**c**) DWSC; (**d**) MWSC.

**Table 1 entropy-24-00096-t001:** PSNR and SSIM values of the tested algorithms in despeckling of simulated SAR images.

Image	Algorithm	L = 4		L = 16	
		PSNR	SSIM	PSNR	SSIM
Lena	Noise	17.7987	0.2643	23.7617	0.4713
	Log-KSVD	23.0580	0.4406	29.7223	0.7523
	Gamma-MAP	25.9796	0.7226	29.7565	0.8146
	PPB POTDF	29.8550 29.6280	0.8025 0.8371	33.2598 34.0313	0.8786 0.8929
	SAR-BM3D FANS	31.1636 31.2764	0.8452 0.8514	34.1672 **34.3950**	**0.8984** 0.8981
	MWSC	**31.4900**	**0.8564**	34.1283	0.8949
House	Noise	17.0168	0.2287	22.9988	0.4362
	Log-KSVD	23.7837	0.5242	28.1420	0.7163
	Gamma-MAP	24.3881	0.6733	28.0729	0.7750
	PPB POTDF	29.6706 28.8839	0.7874 0.8244	33.2822 33.8582	0.8632 0.8746
	SAR-BM3D FANS	31.2560 31.1984	0.8355 0.8415	34.4218 34.3353	0.8886 0.8782
	MWSC	**31.5566**	**0.8469**	**34.5000**	**0.8914**
Cameraman	Noise	17.7353	0.4095	23.7319	0.5629
	Log-KSVD	24.0267	0.6241	28.9515	0.7891
	Gamma-MAP	24.0110	0.7230	28.2264	0.8034
	PPB POTDF	26.9156 27.6073	0.7840 0.8262	29.1476 31.6093	0.8582 0.9023
	SAR-BM3D FANS	28.0520 28.0336	0.8359 0.8384	31.5177 31.7052	0.9086 0.9061
	MWSC	**28.1334**	**0.8424**	**31.7115**	**0.9094**

**Table 2 entropy-24-00096-t002:** ENL and EPI values of the tested algorithms in despeckling of real SAR images.

Image	Algorithm	ENL		EPI
		Region 1	Region 2	
SAR1	**Noise**	9.2121	10.2718	1
	Log-KSVD	83.6841	106.0872	0.3068
	Gamma-MAP	127.8704	188.9452	0.1149
	PPB POTDF	**246.0215** 78.3594	**358.4988** 92.3279	0.1385 0.3227
	SAR-BM3D FANS	64.1340 102.4533	86.2054 178.5487	**0.3520** 0.3444
	MWSC	173.1734	219.8400	0.3516
SAR2	Noise	14.0374	11.6972	1
	Log-KSVD	252.7677	151.5827	**0.4847**
	Gamma-MAP	295.5257	226.2869	0.2924
	PPB POTDF	1366.9521 436.5643	**2939.8065** 549.5423	0.1880 0.3233
	SAR-BM3D FANS	**1470.8629** 554.9834	727.2407 697.4763	0.3109 0.3151
	MWSC	1006.4874	980.5904	0.3134
SAR3	Noise	17.4150	20.3698	1
	Log-KSVD	128.5206	184.8684	0.2989
	Gamma-MAP	149.3338	222.5550	0.2155
	PPB POTDF	**488.0467** 144.8686	**809.7066** 196.7193	0.2472 0.2902
	SAR-BM3D FANS	119.7533 156.1875	159.0166 269.8643	**0.3991** 0.3538
	MWSC	186.0285	306.6940	0.3787

**Table 3 entropy-24-00096-t003:** PSNR and SSIM values of the tested algorithms in despeckling of House Image.

Image	Algorithm	L = 4		L = 16	
		PSNR	SSIM	PSNR	SSIM
House	Noise	17.0168	0.2287	22.9988	0.4362
	Log-KSVD	23.7837	0.5242	28.1420	0.7163
	QWSC	27.3545	0.8284	32.2659	0.8662
	DWSC	27.5353	0.7172	32.3059	0.8597
	MWSC	**31.5566**	**0.8469**	**34.5000**	**0.8914**

**Table 4 entropy-24-00096-t004:** ENL and EPI values of the tested algorithms in despeckling of real SAR2 Image.

Image	Algorithm	ENL		EPI
		Region 1	Region 2	
SAR2	Noise	14.0374	11.6972	1
	Log-KSVD	252.7677	151.5827	0.4847
	QWSC	591.7327	711.9711	0.2075
	DWSC	**1527.7978**	**1029.4798**	0.2541
	MWSC	1006.4874	980.5904	**0.3134**

## Data Availability

Not applicable.

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
