# Peer review of "Synthetic Aperture Radar Image Despeckling Based on Multi-Weighted Sparse Coding"

_entropy, 2022, doi:10.3390/e24010096_

Round 1

Reviewer 1 Report

In general, the paper is very interesting. I’ve only a few comments:

1) Some expressions like “impressive results” (and others similar) seem to be inadequate in an objective discussion. Please, avoid them.

2) The “Cameraman” images are lost. Nevertheless, their results (PSNR and SSIM) are included the table 1. Please, include the images or eliminate “Cameraman” from the paper

3) In order to highlight the behaviour of MWSC, I believe that it is necessary to define a quality factor that combines EPI and ENL. 

4) I think that some comments about patch size and overlapping length are necessary

5) line 146 is a bit confusing:  “y. ||c||”

6) Line 340. How do you define the end of the convergence? What means reach “the tolerance” In figure 2 it seems you have defined M (j>=M) condition. I’m a bit confused about it

Author Response

Details of our amendments are attached.

Reviewer 2 Report

Review of:

Synthetic Aperture Radar Image Despeckling Based on Multi-2 weighted Sparse Coding

This paper introduced an interested application with radar images, however, there are some minor points should be addressed as follows:

Introduction

  1. It is not recommended to show a comparison with very old works in 1980’s. As shown in line 34.
  2. The abbreviation (FANS) was mentioned before it was declared. Line 84.

The proposed SAR despeckling algorithm

Weighted sparse representation model

  1. It should be numbered as (2) not (1). Line 156.
  2. It might be explained how the groups of patches were generated. In line 162.
  3. In fig1, it is not very clear that the first patch is exemplary, and the other patches are similar in both group A and B. line 180.
  4. It might be referred that equation 7 was derived from Gaussian Naïve Bayer. Line 207.

Adaptive matrix parameter learning

  1. Again, it should be numbered properly, for instance here it should be (1.2). Line 243.
  2. It seems that the control factor is vary from image to other image, could you explain this variation in equation 11. Line 248.

Model optimization

  1. Again, it should be numbered properly. Line 275.

Experimental results and analysis

  1. It should mention a reference to authors’ website. Line 351.
  2. In which base or criteria, the size of image patch 8x8 was selected? Line 353.

Despeckling results of simulated SAR images

  1. For Figures 3 to 6, the letter captions were not properly positioned.
  2. Log-KSVD and Gamma-MAP algorithms were omitted because their visual effects were poor. This is clear but why? Line 419.

Despeckling results of real SAR images

  1. Again, it should be numbered properly. Line 441.
  2. For Figures 9 to 13, the letter captions were not properly positioned.

Impacts of weights and dictionaries on performance

  1. Again, it should be numbered properly. Line 509.
  2. For Figures 16, the letter captions were not properly positioned.

Author Response

Details of our amendments are attached.
